# Octodrine: New Questions and Challenges in Sport Supplements

**DOI:** 10.3390/brainsci8020034

**Published:** 2018-02-20

**Authors:** Valeria Catalani, Mariya Prilutskaya, Ahmed Al-Imam, Shanna Marrinan, Yasmine Elgharably, Mire Zloh, Giovanni Martinotti, Robert Chilcott, Ornella Corazza

**Affiliations:** 1Research Centre for Topical Drug Delivery and Toxicology, University of Hertfordshire, Herts SP9 11FA, UK; v.catalani@herts.ac.uk (V.C.); R.chilcott@herts.ac.uk (R.C.); 2Department of Pharmacy, Pharmacology and Clinical Science, University of Hertfordshire, Herts AL10 9AB, UK; m.zloh@herts.ac.uk; 3Semey State Medical University, Republican Scientific and Practical Center of Mental Health, Pavlodar 140002, Kazakhstan; mariyapril2407@gmail.com; 4Faculty of Medicine, University of Baghdad, Baghdad 10071, Iraq; tesla1452@gmail.com; 5Parliamentary Office of Science and Technology, Houses of Parliament, London SW1A 0AA, UK; marrinans@parliament.uk; 6Navy General Hospital, Cardiovascular department, Alexandria 21513, Egypt; yasmine_elgharably@ymail.com; 7Department of Neuroscience, Imaging and Clinical Sciences, “G.d’Annunzio” University, 66100 Chieti, Italy; Giovanni.Martinotti@gmail.com

**Keywords:** octodrine, dimethylhexylamine, DMHA, ambredin, fitness, novel psychoactive substance, performance and image-enhancing drugs, anti-obesity agents, weight loss

## Abstract

**Background:** Octodrine is the trade name for Dimethylhexylamine (DMHA), a central nervous stimulant that increases the uptake of dopamine and noradrenaline. Originally developed as a nasal decongestant in the 1950’s, it has recently been re-introduced on the market as a pre-workout and ‘fat-burner’ product but its use remains unregulated. Our work provides the first observational cross-sectional analytic study on Octodrine as a new drug trend and its associated harms after a gap spanning seven decades. **Methods:** A comprehensive multilingual assessment of literature, websites, drug fora and other online resources was carried out with no time restriction in English, German, Russian and Arabic. Keywords included Octodrine’s synonyms and chemical isomers. **Results:** Only five relevant publications emerged from the literature search, with most of the available data on body building websites and fora. Since 2015, Octodrine has been advertised online as “the next big thing” and “the god of stimulants,” with captivating marketing strategies directed at athletes and a wider cohort of users. Reported side-effects include hypertension, dyspnoea and hyperthermia. **Conclusions:** The uncontrolled use of Octodrine, its physiological and psychoactive effects raise serious health implications with possible impact on athletes and doping practices. This new phenomenon needs to be thoroughly studied and monitored.

## 1. Introduction

The evolution of trends within drug use has recently been marked by a rapid expansion in the number of commercially-available psychoactive substances [1], with an increased number of young users [2] and relevant psychiatric consequences [3]. This includes both a proliferation of new drugs (‘research chemicals’ or ‘RC’s) with a distinct pharmacology and very little associated research evidence on their physiological or side effects, as well as an increase in the abuse of diverted prescription medications [4,5] Octodrine sits somewhere between these two trends, being a traditionally-developed pharmaceutical but with no current, legitimate medical application.

The so-called “Performance and Image-Enhancing Drugs” (PIEDs) taken to enhance human abilities in a myriad of spheres, are one important emerging facet within this. These include substances with a perceived ability to enhance physical performance, psychological status, appearance, cognitive abilities and social relations and as such are sometimes referred to as ‘lifestyle drugs’ [6,7,8,9,10]. The concept of PIEDs is now well established and is acknowledged particularly in relation to the world of athletics [11,12]. The most well-known PIEDs are the anabolic steroids, peptides and hormones but their use is increasingly giving way to other types of substance to achieve specific goals. These can be physical in nature (e.g., tanning, weight loss, muscle gain, speed, strength, performance) or cognitive, such as the use of nootropics for professional or academic performance [13,14], or for social gain, where various categories of substance as a ‘social lubricant’ for social anxiety support). Over the past decade, more than 800 NPS were identified in over 102 countries by the EMCDDA and the UNODC Early Warning Systems [15,16] as well as our ongoing monitoring activities [1] and their number is constantly growing. Some of these compounds may represent a serious issue for public health and are changing the face of debates around doping by playing unfairly on the narrow line between legal and illegal [12]. The globalization of the online drugs market has made this a widespread phenomenon, reaching a new cohort of users, which includes not only the body builders and time-pressured professionals, who were initially associated with this trend but also students and others of all demographics [12,17,18]. 

In November 2016, Octodrine was found in an athlete engaged in a bodybuilding competition, later disqualified as he also tested positive for anabolic and stimulant drugs, included in the World Anti-doping Agency's (WADA) List of Prohibited Substances (Section S6 and S1) [19,20]. Octodrine is a psychoactive central nervous system stimulant. It is an amphipathic primary amine (Figure 1) [21] known under many names, including dimethyl hexylamine (DMHA) and 2-amino-6-methylheptane, 2-metil-5-amino-eptano. Its structure presents some similarities with that of other illegal stimulants like, AMP Citrate (DMBA), Ephedrine and 1.3-DMMA itself. With DMAA and AMP Citrate already phasing or phased out of current supplements, this drug was brought back on market as an alternative in pre-workout and ‘fat-burner’ products in 2016. Octrodrine was originally developed in the United States as an aerosolized treatment for bronchitis, laryngitis and other conditions [22,23,24]. Its pharmacology was studied in the early 1950s, was investigated as an antitumor drug and used to be available as a nasal decongestant under the tradenames Vaporpac and Tickle Tackel Inhaler [25]. Sympathomimetic effects of DMHA were explained as alpha adrenergic agonist-mediated via G-protein-coupled receptors (GPCRs) [26]. Limited human data is available just from preliminary studies, while studies on activity and acute toxicity had been conducted on animals (cats, rabbits, dogs and pigs) [27,28,29,30,31,32]. Octodrine was found to increase the pain threshold, cardiac rate (positive chronotropic effect) and myocardial contractility (positive inotropic effect) [33,34,35]. The safety of Octodrine as an individual drug remains unknown due to the lack of any placebo-controlled trial but animal experiments suggest a potential for adverse cardiovascular effects. Structurally, there are two forms of DMHA: the naturally occurring 2-amino-5methylpetane and the synthetically derived 2-amino-6-methylheptane. The natural version can be found in extracts of Juglans Regia (Walnut Bark), Aconitum Kusnezoffii’s and Kigelia Africana and it is often used for hunting purposes [36,37,38,39,40,41,42,43]. The synthetic version is the most widely used because less expensive and toxic to produce. It is therefore assumed that the DMHA used in supplements is synthetic. As of right now, this molecule is not on the 2016 WADA banned substances list but it fits perfectly in the category of the well-known Performance and Image Enhancing Drugs (PIEDs). Coveted by elite track and field athletes, DMHA is marketed to a broader demographic including beginners and non-professionals.

Considering the existing knowledge gap spanning seven decades and the re-emergence of Octodrine as a new drug trend, it was felt the need to further investigate the phenomenon in different communities, while exploring issues related to its e-commerce, consumption, motivations of use and potential negative impacts to health, among other features.

## 2. Materials and Methods

A literature review on Octodrine was carried out in the following databases: Scopus, Medline, EBSCO and Google Scholar (Figure 2). A list of keywords was compiled in accordance with a preliminary pilot study of literature and databases on the surface web and online e-commerce websites. Terms included: “Octodrine,” “Ambredin & Vaporpac,” “2-aminoisoheptane,” “Dimethylhexylamine,” “DMHA,” “2-amino-6-methylheptane,” “6-methyl-2-heptylamine,” “2-metil-5-amino-eptano,” “5-methyl-2-heptylamine,””Dimethylhexylamine,” “Aconitum kusnezoffii, “Aconite extract,” among others. The keywords also included synonyms of Octodrine in other languages and names of chemical isomers. Searches were carried out in English, Italian, German, Arabic and Russian.

No time restrictions were applied to the searches. Inclusion and exclusion criteria for literature data selection are defined in Table 1. Considering the lack of scientific investigations in the field and the absence of experimental and/or interventional studies in humans, additional qualitative systematic searches were carried out in the world-wide web to investigate the extent of diffusion of Octodrine, trading strategies for its distribution and the nature of the self-reported (subjective) experiences by users in English, German, Arabic and Russian. These included bodybuilding websites, chemistry and chemists’ websites, pharmaceutical companies, online e-commerce stores as well as a range of fora posts/threads. The web snapshot was carried out on a regular basis (between November 2016 - January 2018) using a Google search. Only publicity available information was considered for the study and no posts/other contributions to fora discussions were made by the researchers. Additional data were also obtained by consulting Google Trends [44].

Ethical approval for this the study was granted by the School of Pharmacy Ethics Committee, University of Hertfordshire, Hatfield, United Kingdom (November 2013; PHAEC/10-42). 

## 3. Results

### 3.1. Medical and Paramedical Database, Grey Literature

Various articles emerged from our literature searching but only eight of them [23,24,27,28,29,30,31,33] referred to Octodrine, Octodrine derivatives and Octodrine-related compounds in the entire scholarly-published literature (Table 2). 

Reference to its multiple medicinal properties was found in five of these papers, which highlighted its sympathomimetic and broncho-spasmolytic effects, with possible further actions as a stimulant, anti-obesity and appetite suppressant agent. The molecule is cited also as an antimicrobial with specific antifungal activity [45], as a nasal decongestant [47] and as an ingredient of dietary supplements [48]. Other scholarly papers (a total of seven) made passing or limited reference to Octodrine, covering the chemical properties and analyses of several compounds including this one, or providing data on its antimicrobial effects only [45,46]. These are other scattered examples of relevant documentation, including an invention patent from 2012 [49]. However, the lack of experimental randomized controlled trials (RCTs) and other interventional studies on humans has led to a complete absence of systematic reviews and meta-analytic studies related to use of Octodrine as a medicinal agent or food supplement. Two of the three papers found on PubMed were published in the *Journal of Pharmacology and Experimental Therapeutics* in 1947 and 1951 respectively [27,30], while the third paper [29] was published at the *Archives internationales de pharmacodynamie et de thérapie*. Since the 1950s, there have been no other scholarly-published data specific for Octodrine in any peer-reviewed journal, neither observational nor experimental could be found on the entire web, including medical and paramedical databases, or unpublished literature. The substance remerged on the literature in 2017, when Cohen et al. published a study conducted on six different supplements: Game Day, Infrared, 2-Aminoisoheptane, Simply Skinny Pollen, Cannibal Ferox AMPed and Triple X. All these products disclosed on their label the words Octodrine, 2-amino-6-methylheptane and 6-methyl-2-heptanamine or listed the stimulant as if it were an extract of Aconitum kusnezoffii plant. Results showed that only one of them, Game Day, contained Octodrine, while the others contained different or banned stimulants [50].

#### 3.1.1. Limited Data-Reporting in Scholarly Peer-Reviewed Papers and Invention Patents

There is limited mention of Octodrine in invention patents from 2012 in relation to a novel stable anaesthetic for reducing skin reactions [49].Two papers, pertinent to the disciplines of toxicology and chemical chromatography, examined Octodrine in terms of its physiochemical properties including relative retention time (RTT) and its identification in hair samples [51,52]. Furthermore, Niu et al. and Kim et al. [45,47] discussed the broad-spectrum antimicrobial effect, antifungal effect, anti-persister activity and application for the treatment of Candida albicans and uropathogenic strains of Escherichia coli. These two papers also discussed Octodrine microbial resistance. Additionally, Kuo et al. (2004) [26] and Schlessinger et al. (2011) [53], documented the sympathomimetic properties of Octodrine and effects related to norepinephrine transporter (NET) and G-protein-coupled receptors (GPCRs), which was in concordance with the results from 1947 and 1951 animal studies [27,30].

#### 3.1.2. Google Trends

Google Trends provided valuable data in relation to the interest in Octodrine on the Web. Four keywords provided good insight on the trend as far back as the year 2004. These keywords are “Octodrine,” “2-aminoisoheptane,” “aminoisoheptane” and “DMHA.” There was an obvious incremental increase of interest in Octodrine starting in the year 2012. This interest plateaued between 2013 and 2014 and was followed by a steep rise in 2014–2015, followed by a further escalation starting in 2015 before peaking by the January of 2018 [54]. Comparing to other three keywords, DMHA has demonstrated the greatest interest among Google users ranging between 9 searches in July 2008 and 100 searches in September 2017. The leading countries in terms of internet searches of Octodrine (DMHA) were the USA, Canada and Australia. On the Russian-language Internet, users showed no search activity for Octodrine, while intensively searching for DMAA. In June 2017, the quantity of DMAA searches in Russian-language zone was 100. Since 2004 the trend has demonstrated stable growth in this local online area [55]

#### 3.1.3. Bodybuilding Website, (Bio)Chemistry, Pharmaceutical Websites, Blogs and Online Fora

Body building websites provided a major source of data, especially in relation to the analysis of online trading platforms and fora discussing Octodrine and its effects. A multilingual approach used in this part of the study facilitated the characterisation of regional and national features of sport-stimulant markets. 

No specific inclusion criteria were imposed upon the body-building websites, beyond demonstrating a mention of DMHA or synonym. All such instances were included in the evaluation. 

The English-language domain was investigated with relevant results. No results were produced from searches in Arabic. Thousands of websites can be located using the Google search terms “Octodrine” and its synonyms [56,57,58,59]. Popular brand names include: Olympus Labs CONQU3R Unleashed, Total War, Simply Skinny Pollen, AdrenaCLENV2, Game Day, Cannibal Ferox Amped, Giant Sports Giant Rush [57,58,59,60,61,62,63,64,65,66]. By January 2018, 68 English- and 6 German-language online shops selling Octodrine were identified. The product is often advertised as the “next big thing” in bodybuilding environments and described as the “new MDAA” whose effects are “just right” for dietary supplement users and/or stimulant-enthusiasts as it can allegedly enhance focus, experience and performance. Many of these sites also provide detailed information around usage and dosage, alongside with warnings on risks and severe side effects of this emerging molecule [56,57,58,59]. Professional scientific or pharmaceutical sites regarding the chemistry characterization of this compound can be found on the web, as well as “amateur” websites, displaying more generic scientific information on this potentially dangerous substance [67,68]. Online stores are predominantly American or Australian domains that ship their products all over the world. Octodrine is presented as a DMAA-like stimulant and predominantly sold as a fat-burner product or pre-workout formula. Some websites also recommended it for intensive study sessions, positioning this molecule among the Nootropics; pharmaceuticals used to improve cognitive and executive function, memory and creativity in healthy individuals. Claims such as “It boosts dopamine and noradrenaline uptake, while slowing down reuptake just long enough for a solid workout or study session” are quite common [58]. 

Usually Octodrine is sold in powder (e.g., Cannibal Ferox AMPed, Olympus Labs CONQU3R Unleashed, Game Day, Total War) as pre-workout or in capsules (Infrared, Simply Skinny Pollen) as fat burner, with prices ranging from 1.75 to 3.75 dollars per serving.

German online trading platforms focus customers’ attention on Octodrine with detailed feedback from the reviewers (estimation of taste, effects, “price-quality”) [68] or vague offers of “fitness hardcore pre-workout booster for pumps and focus” [69]. The trader for amazon.de mentions Octodrine as additives “Oct” in “Arginine AKG, Beta-Alanine, Citrullin Malate complex” and omits a description of its side-effects. Standard marketing technologies widely used by German traders are also employed, such as discounts, world-wide express-delivery and even “halal” certification. 

In terms of the Russian-language results, the majority consisted of bodybuilding resources linked to 73 online shops delivering Octodrine to the Russian Federation, the Ukraine and 4 Central Asian countries. In contrast, fora did not indicate significant popularity among local athletes and were predominantly arranged within Russian social nets [70,71]. All identified Russian-language websites run their activity legally on the surface web, a fact explained by the absence of any law enforcement restrictions referring to DMHA and DMAA in Russia and Ukraine. The relative lesser popularity of these particular stimulants is no doubt influenced by the availability of much cheaper locally produced analogues. An imported stimulant complex with DMHA and DMAA (from the USA) costs 1.27 ± 0.19 USD per unit compared to a local analogue for 0.49 ± 0.23 USD per unit. The local online platforms offer more than 20 brands of Octodrine. The immense variety of trading names of this substances is attributable to continuous rebranding as attempts to overcome counterfeit production [72]. By offering athletic stimulant complexes, Russian online shops strive to advertise DMHA and DMAA as the active components for the desirable results; less attention is paid to other substances such as vitamins, tyrosine, taurine and DMAE [73,74]. Trading and producing companies announced anti-inflammatory, anaesthetic, spasmolytic and anticonvulsant properties of Octodrine offering the “ideal” substance for “hard-core” training [72]. The use of Octodrine as a weight loss product is rarely advertised (e.g., only 9 Russian-language online shops were identified). Fat burners containing Octodrine are typically sold at a higher price point: 1.3 USD per unit in contrast to the 0.5 USD for the pre-training complex. Taking into account the legality of DMHA and DMAA in Russian-language territories and the absence of relevant trading regulations, local online shops did not notify their customers to possible side and toxic effects of the stimulants. Some shops explicitly claim that Octodrine has no side effects and that is potential is the same as caffeine [75]. For example, berserktakticalfarma.blogspot.com underlined the safety and high effectiveness of DMHA, emphasising that its potency is equal to 90% of DMAA [76]. Only one of 73 identified online websites warned that Octodrine and DMAA can be detected and could possibly mislead sport competition testing and thus advised users to cease Octodrine beforehand [74]. An additional online shop mentioned contraindications generally for stimulant complexes without specification on Octodrine [77]. 

Discussions on fora include suggested dosages, combinations, duration of action, among others. According to such anecdotal evidence, a “safe dose” is considered to be around 1mg/kg of bodyweight up to 160 mg per day, while others recommend 100 mg of the synthetic DMHA isomer and 75 mg of the natural one to reach the “sweet spot.” In the Russian-language internet zone, the recommended dose of DMHA substantially differed between online shops (30 to 400 mg), as did the dosing schedule. For instance, the online retailer hulkfood.ru advised to take Octodrine for 45 days without stopping to gain significant desired stimulating effects before sporting competition [78]. 

Users suggest an intake approximately 15–30 or 30–60 min prior to working out [79,80]. Alternatively, if used for its appetite suppressant properties, DMHA consumption was advised between meals and never in the evening as it might affect sleeping. According to users, 25 mg twice a day are enough to “keep one’s mind off food” [81]. Experienced users shared their insights: “If used predominantly for its appetite suppressant properties, DMHA can be used during the day between meals. However, we recommend taking caution if using this ingredient late in the afternoon or early evening, as it has the potential to hinder your ability to sleep.” 

DMHA effects will occur ~15 to 60 min after consumption. The substance demonstrates potency to heighten level of mental focus, increase energy and reduce appetite, as well as raise feelings of wellbeing. Bloggers described a three-phase effect for pre-work-out complexes containing DMHA: (1) stimulation, (2) post-stimulation side effect symptoms and (3) sleep disturbances. Octodrine is frequently advised as a substance intensifying the first two phases with no impact on sleep [82]. Because of its stimulants effects, Octodrine has also been used outside fitness settings, including working environments as non-prescribed medication [83,84]. To boost athletic performance, it is sometimes ingested in combination with huperzine A, DMAE, n-acetyl tyrosine, alpha-gpc, noopept, phenotropil and picamilon to gain maximal focusing on training. Phenibut and ladasten strengthen the euphoric effects of Octodrine [82]. 

Side effects such as mood swings, tremor, concentration deficiency, over-stimulation, energy crashes, anxiety, high blood pressure, dyspnoea, rapid heartbeat and heartburn have been reported 6–8 h after the initial onset of effects [85,86]. Some users also reported eyes twitching (blepharospasm), pulsing sinus area (carotid sinus), mood fluctuation, absent-mindedness; a rise in blood pressure, piloroerection and hyperthermia following ingestion of Octodrine (e.g., Anabolicminds.com, 2016; Project Bodybuilding, 2016; kandeleria.ru). There were only a limited number of indications for professional athletes: some websites suggest avoiding consumption for ethical considerations, or because it could be considered a violation of the WADA restrictions [72,86,87].

Some fora provide information on abuse potential and possibility of dependency on Octodrine. Namely, users warned about withdrawal symptoms and growth of tolerance resulting from the short- and long-term use of the stimulant [71,82]. Experienced athletes recommend alternating 3-4 stimulant complexes each training day to overcome undesirable adaptation [77].

## 4. Discussion and Conclusions

To the best of the authors’ knowledge, this study is the first to implement a systematic review of the literature or undertake cross-sectional analysis of the content of the web in relation to the diffusion and e-commerce of Octodrine. The authors also bring to the spotlight the noteworthiness of subsequent studies on Octodrine, specifically chemical analysis and receptor-ligand binding assays, with the aim is of reaching a full understanding of this widely used substance. The restrictions on amphetamine-type stimulants (e.g., DMAA, DMAE) exacerbate demands on new formulas tailored to the ambitions of bodybuilders and athletes. The sympathomimetic properties of Octodrine, inheriting the potential of DMAA, meet the expectations of “new-generation boosting” and stimulate the growth of global trading. Meanwhile discrepancies between practical experience and theoretical knowledge were observed in our study. Very little is known about its pharmacodynamics and pharmacokinetics profiles, or the chemical profile of the commercially-available Octodrine-related products. 

On the Internet, Octodrine is misleadingly advertised as a “safe and legal” analogue of banned stimulants (e.g., DMAA and phenethylamine), making it potentially more attractive to new and experienced users. Intensive marketing campaigns with unlimited worldwide delivery, discounting programs and displays of “good” feedback from reviewers, have contributed to the re-emergence and current spread of Octodrine in the drug market. According to anecdotal for a, reports and trading information, motives of use go beyond athletics gyms: more often the drug is recommended as a day-life stimulator.

The psychoactive effects of Octodrine were neither previously described in literature nor studied despite its structural similarity to other drugs of abuse (DMAA, DMAE). Its metabolic pathway and adverse reactions have not been studied in humans, making its use in fitness settings extremely hazardous. A limited number of open claims were found on the potential risk, side effects and complications of Octodrine use, while its properties as “hard-core” for advanced users are often emphasized. Sites exploited fragmented and sporadic “scientific news” describing only favourable effects of Octodrine, with research evidence being completely omitted. Only some English and German sites warned about ceasing Octodrine use before sport competitions. Meanwhile, on Russian-language trading platforms, DMHA was actively offered in combination with DMAA - attributable to the legal status of both in this geographical zone (Russia, Belarus, Kazakhstan, Kyrgyzstan). Amphetamine-type stimulants (such as DMAA) have showed more popularity, especially for the last two years. 

The reported side effects of Octodrine suggest the strong need for further research. The desirable, stimulatory effects of Octodrine are accompanied with a range of mental and somatic symptoms, with frequent use of Octodrine being associated with tolerance, withdrawal symptoms and risks of dependence syndrome. We reiterate here the importance of focusing and asking direct questions on the pharmacological and toxicological properties of Octodrine. The addictive potential should be promptly identified and assessed. Hence, attention should be paid to further investigation in the field and consider the incorporation of Octodrine within WADA and FDA prohibited substances lists. 

There are some limitations to our study. Only five peer-reviewed papers were directly pertinent to Octodrine and these were mainly published 6–7 decades ago. Other scholarly-published studies addressed Octodrine in a marginal way. The paucity of the available literature, absence of review articles, systematic reviews and meta-analytical studies clearly limit the extent of this present review. Taking into consideration lack of publications on the chemical analysis of Octodrine, a comprehensive chemical analysis is necessary for future research, with the aim of establishing the detection and identification of Octodrine, along with potential contaminants, excipients and other active ingredients in the currently promoted powder-products under the name Octodrine or DMHA. The sympathomimetic effect of Octodrine is well-understood. However, the central effects including the psychostimulant and anti-obesity effects are not adequately explored and/or reported, with no formally documented experimental studies nor case reports. Future studies should focus on the central effect of Octodrine and its correlation with patterns of cerebral dominance and the lateralization of brain function. Moreover, there is an inadequate body of data in relation to the geographic usage of Octodrine, particularly for contrasting the developing world (including the Middle East, North Africa and post-soviet regions) versus the developed world. In terms of multilingual analysis of websites and drug fora, only publicity available information was considered for the study to uphold observational status. Fora requesting registration were not included in the study. Self-reported experiences are only partially reliable and it may be inappropriate to trust such anecdotal evidence without independent verification.

## Figures and Tables

**Figure 1 brainsci-08-00034-f001:**
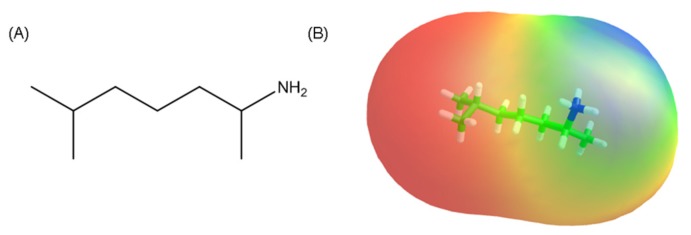
Chemical structure (**A**) and molecular lipophilicity potential (MLP) surface (**B**) of octodrine molecule (hydrophobic surfaces are depicted in red and polar surfaces are in blue) [21].

**Figure 2 brainsci-08-00034-f002:**
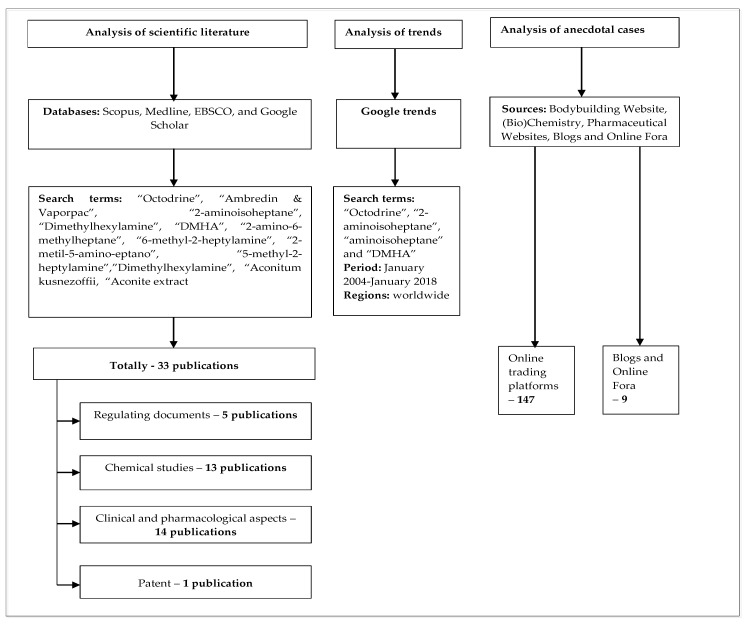
Algorithm of the analysis.

**Table 1 brainsci-08-00034-t001:** Inclusion and exclusion criteria for selection of articles and the web analysed in this study.

**Inclusion Criteria**
Studies and publication related to Octodrine Studies and publication of octodrine-related compounds and chemicals, in which Octodrine is an ingredient Studies and publication in which octodrine is marginally included English, German and Russian languages All years of publication (no date restriction) Surface web Grey (unpublished) literature, including master’s and doctorate theses Fitness and body building websites (Bio)chemistry, pharmacy and pharmaceutical websites Online drug fora Human and animal studies Observational and experimental studies
**Exclusion Criteria**
Duplicate Articles Initial screening for relevance (reading the title and abstract) Articles found to be irrelevant by analysing the full article Low scoring for an article on CASP critical appraisal tool (poor quality of appraised manuscript)

**Table 2 brainsci-08-00034-t002:** Pharmacological and clinical properties of Octodrine (analysis of articles).

Reference	Author	Year of Publication	Name of Studied Substance or Medicament	Key Findings
Respiratory system				
[28]	Charlier, R.; Philippot, E.	1950	theophylline-diethylenediamine ethanoate	The aerosol with Octodrine demonstrated the property to increase respiratory volume
[29]	Charlier, R.	1951	2-amino-6-methyl-heptane	Animal experiment (dog) revealed bronchodilation, increased nasal and lung volume caused by 2-amino-6-methyl-heptane
[23]	Gode, J.	1958	Ambredin	Identification of bronchospasmolitic properties of Ambredin medicament consisting of Aceverine Hydrochloride, Octodrine Phosphate and Theophylline
[24]	Tschudin, M.L.	1960	Ambredin	
Cardiovascular system				
[30]	Fellows, E.J.	1947	2-amino-6-methylheptane	2-amino-6-methylheptane hydrochloride caused an increase in cardiac rate and amplitude of contraction in animal experiment (dog)
[27]	Marsh, D.F.; Herring, D.A.	1951	Methyl-2-heptylamine	Compared to others sympathomimetic amines, 6-Methyl-2-heptylamine focused the myocardial stimulant activity and increased force of myocardial contraction along with heart rate
[29]	Charlier, R.	1951	2-amino-6-methyl-heptane	Animal experiment (with dog) revealed growth in arterial blood pressure after the exposure of 2-amino-6-methyl-heptane
[34]	Oelkers, H.A.	1967	2-amino-6-methylheptane (+)-camphor-10-sulfonate	Inotropic properties of 2-amino-6-methylheptane (+)-camphor-10-sulfonate were identified
[31]	Trieb, G.; Nusser, E.	1974	Ordinal® retard	The medicament Ordinal® retard combining Octodrine, 3-octopamine and adenosine demonstrated pressure effects in treatment of patients with hypotension
Nervous system				
[30]	Fellows, E.J.	1947	2-amino-6-methylheptane	2-amino-6-methylheptane demonstrated local anaesthesia and elevation of local pain threshold in experiments with animals (rabbits, cats, dogs)
*Antimicrobial activity*				
[45]	Kim, K.; Zilbermintz, L.; Martchenko, M.	2015	Octodrine	Octodrine demonstrated antifungal activity in experiments with serum-grown *C. albicans*
[46]	Niu, H.; Cui, P.	2015	Octodrine	Octodrine demonstrated experimental activity against stationary phase *E. coli*

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
