# Peer review of "Octodrine: New Questions and Challenges in Sport Supplements"

_brainsci, 2018, doi:10.3390/brainsci8020034_

Reviewer 1 Report

In the manuscript “Octodrine: New Questions and Challenges in Sport Supplements” the authors did observational & cross-sectional analytic study of Octodrine and its usage related health concerns. Authors collected data from several internet sources and peer-reviewed publications.  The study was nicely performed, presented and gives us warning signals for the usage or abuse of Octodrine. Some of the minor concerns are following: 1. Author could present the details of the five peer-reviewed publications used for analysis as a Table. 2. The abbreviation of DMHA should be consistent throughout the manuscript. 3. If the authors include, a schematic /graphical representation summarizing the data search, analysis and findings of the study, it would greatly increase the readability and understandability

Author Response

Dear Reviewer,

Thank you for your comments. Following your suggestions:

1.       A table (Table. 2 in the text) has been inserted with the pharmacological and clinical properties of Octodrine emerged by the analysis of the peer-reviewed articles.

2.    In response to this comment we searched the text for other synonym of the word DMHA. We found only DMHA and Octodrine as words used to refer to this molecule. In our opinion both could be used as synonym in reference to the same substance along the text, considering also the high use of Octodrine word

3.     A graphical representation that summarize the data search, analysis and finding of the study has been included in the paper as Fig. 2

Best Regards

Reviewer 2 Report

The authors have presented a very interesting systematic review about octodrine use as a sport supplement. This study sheds light on the importance of initiating well designed observational and interventional studies on the use of octodrine as a supplement. Due to lack of well represented literature on this subject, it would have been difficult to come up with specific recommendations out of the results. I have some minor comments to take into consideration

1-          I found the abstract conclusion to be weak. The authors described the results of their systematic search. However, did not mention any statements in the abstract conclusion about the results. A more detailed conclusion that reflects the results will give the study higher strength and value as it can be used as a reference for other researchers interested in this topic.

2-          In the introduction, the authors mentioned the name of a body builder “Nathan Tait” as an example on athletes that misused drugs. I don’t see any additional value in using the specific name of the athlete in fact it would be more professional not to mention personal names.

3-          The authors mentioned that they have used body building websites to collect information from. The methods section failed to show any specific criteria of choosing which websites to include. This may pose a risk of unintentional selection bias. Could the authors elaborate on this point ?  

Author Response

Dear Reviewer,

Thank you for your comments. Following your suggestions:

1-      The conclusion of the abstract has been changed in order to give more values to the threat that Octodrine, with its  physiological and psychoactive effects, could represent to the public health, and its possible impact on athletes and doping practice

2-      The specific name of the ATHLETE has been removed and changed with a generic term.

3-      Beyond demonstrating a mention of DMHA or synonym, no specific inclusion criteria were imposed upon the body-building websites, and all the website with a positive hit were included

Best Regards